# Residual Stress Distribution in Water Jet Peened Type 304 Stainless Steel

**Makoto Hayashi** [1,*,†], **Shinobu Okido** [2] **and Hiroshi Suzuki** [3]

[1] Hitachi: Ltd., 1-1, Saiwai-cho, 3-chome, Hitachi, Ibaraki 317-0017, Japan

[2] Hitachi-GE Nuclear Energy, 1-1, Saiwai-cho, 3-chome, Hitachi, Ibaraki 317-0017, Japan; shinobu.okido.hz@hitachi.com

[3] Japan Atomic Energy Agency, 2-4, Shirakata, Tokai, Ibaraki 319-1195, Japan; suzuki.hiroshi07@jaea.go.jp

\* Correspondence: zv-hayashimak@aida.co.jp or hayashijapan7@msn.com; Tel.:+81-42-772-5231

† Now at AIDA ENGINEERING, LTD.

**Abstract:** In materials with a surface treatment such as shot peening, the residual stress gradient in the surface layer is severe. When measuring the residual stress distribution near the surface with a severe stress gradient by the neutron diffraction method, the gauge volume must be removed from the measurement sample. However, when the gauge volume deviates from the sample, a pseudo peak shift occurs and accurate stress distribution cannot be evaluated. Therefore, it is necessary to evaluate the pseudo peak shift in advance under the same conditions, as in the case of actual residual stress measurement, using a sample in an unstressed state. In this study, the stress distributions in the surface layer of a type 304 stainless steel plate and bar with simulated stress-corrosion cracks which were subjected to water jet peening—giving a surface layer residual stress equivalent better than that of normal shot peening—were evaluated considering the pseudo peak shift. As a result, the residual stress distributions in the surface layer were measured in good agreement with the measurement result obtained by the sequential polishing X-ray diffraction method. It was clarified that the residual stress distribution in the near surface with steep stress gradient can be evaluated by the neutron diffraction method.

**Keywords:** residual stress; neutron diffraction; water jet peening; pseudo peak shift; stress gradient; austenitic stainless steel

## 1. Introduction

The X-ray diffraction technique widely used to accurately measure residual stress in various kinds of materials has been standardized by the Committee on X-ray Study for Deformation and Fracture of Solid of The Society of Materials Science, Japan [1]. Since the penetration depth of X-ray is only about 20 μm from the surface, residual stresses inside the structure cannot be measured. On the other hand, the penetration depth of a neutron is sufficiently deep and the neutron diffraction technique is the only method available to non-destructively determine residual stresses inside weldments [2,3]. The neutron diffraction technique has also been applied to the stress distribution measurement at the fatigue crack tip of specimens under loaded and unloaded conditions [4]. The authors have examined the residual stress distributions in the structural components and pipes [5–9].

In order to improve the fatigue strength, surface treatments such as shot peening are applied to structural parts and components. At that time, the residual stress gradient in the surface layer is very much severe. When measuring the residual stress distribution near the surface with the severe stress gradient by the neutron diffraction method, the gauge volume must be removed from the measurement sample. However, when the gauge volume deviates from the sample, a pseudo peak

shift occurs and accurate stress distribution cannot be evaluated. Therefore, it is necessary to evaluate the pseudo peak shift in advance under the same conditions as in the case of actual residual stress measurement using a sample in an unstressed state.

In this study, the stress distributions in the surface layer of type 304 stainless steel plate and bar which were subjected to water jet peening, giving the surface layer residual stress equivalent better than that of normal shot peening, were evaluated considering the pseudo peak shift.

## 2. Water Jet Peeing

In order to improve the fatigue strength, steel shot peening is usually applied to structural parts and components in various machines. On the other hand, stress corrosion cracking has been initiated in internal components in pressure vessels of light water nuclear power plants. However shot peening cannot be applied to the internal components at such sites, because the steel shots have to be collected absolutely in order to avoid unexpected damage to the component. Thus, the water jet peening method has been developed [10,11].

The principles of water jet peeing are shown in Figure 1. The water is pressurized up to about 60–70 MPa using high pressure pump. Highly pressurized water is injected into water through water jet nozzle. The diameter of the nozzle is about 2 mm. By the injection of highly pressurized water into the material, an extremely large number of cavities are generated due to boiling decompression. The distance between nozzle tip and the target material is usually longer than 100 mm and depends on the peening conditions. The depth of water when constructing water jet peeing is 1.5 m and the scanning speed of the nozzle is 0.4 m/s.

When the cavity collapses on the material's surface, a very high pressure is generated and the material's surface is plastically deformed. Thus the large compressive residual stress is generated similarly to normal shot peeing.

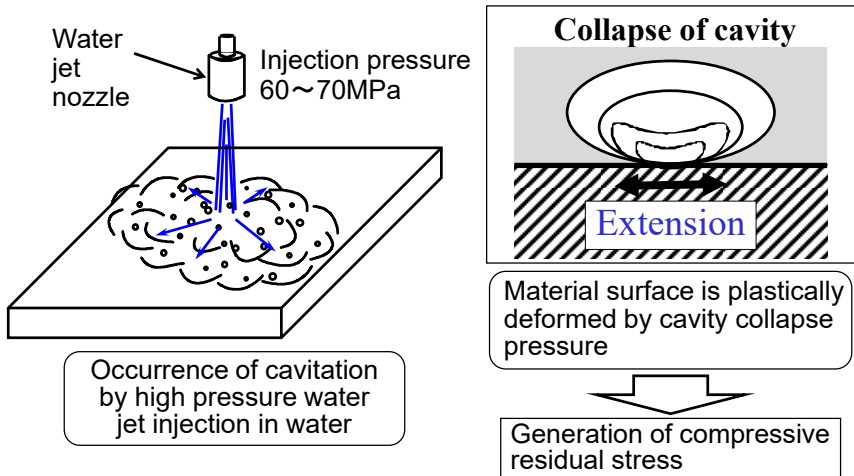

**Figure 1.** Principle of water jet peening.

Water jet peening system is shown in Figure 2. The system is very simple. Water is supplied from pure water tank. It is pressurized to about 60 MPa by a high pressure pump. Pressurized water is transported to the nozzle through a high pressure hose. The nozzle is installed on an XYZ scanner so that the water jet peening is applied to the objective material in horizontal plane and the distance between the nozzle and the material can be controlled. The water flown out from the test chamber returns to the pure water tank.

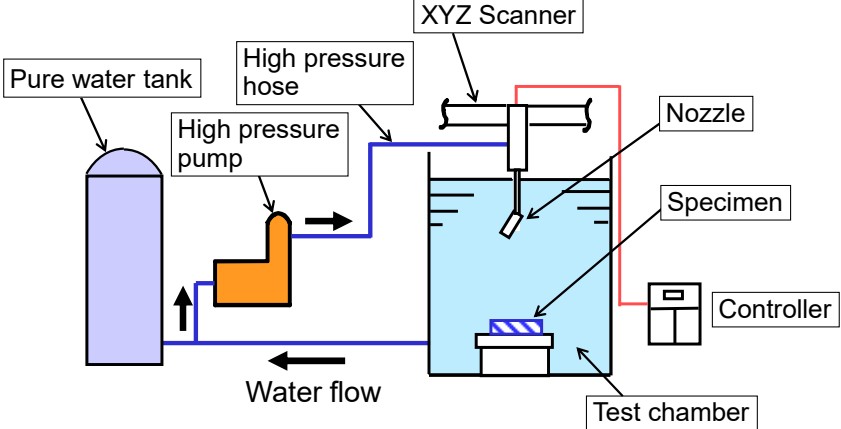

**Figure 2.** Water jet peening system.

The appearance of water jet peening is shown in Figure 3. Highly pressurized water flowing out of the nozzle rapidly expands due to the boiling decompression, and a large number of cavities are generated. In the figure, the water flowing out from the nozzle and the numerous cavities, appear white. It expands gradually and impinge to the material surface over a wide area.

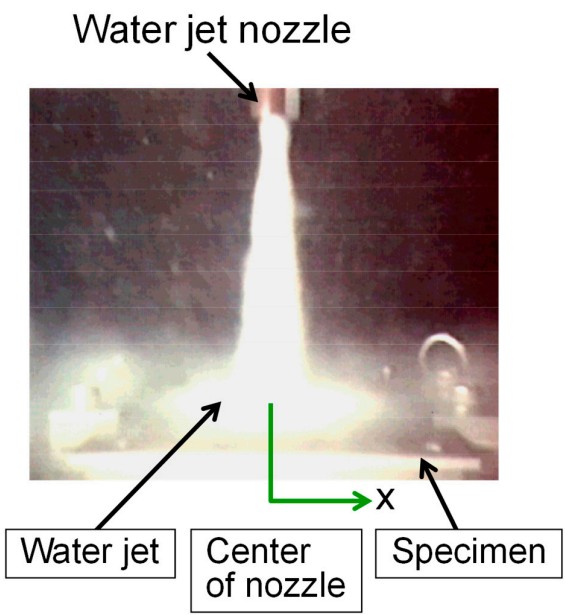

**Figure 3.** The appearance of water jet peening.

The residual stress distribution on type 304 stainless steel plate surface measured by X-ray diffraction is shown in Figure 4. Before water jet peening the residual stress ranges from 400 to 520 MPa due to grinding. After water jet peening the residual stress is about −500 MPa at the center of peening. A very wide area is covered with the very high compressive residual stress. If assuming the effective residual stress as −400 MPa, the effective area is wider than 60 mm. In the case of laser peening, the effective area is very narrow because of the small diameter of the laser beam—about 1 mm [12]. Thus it is found that the water jet peening effectively reduces tensile residual stress to compressive residual stress.

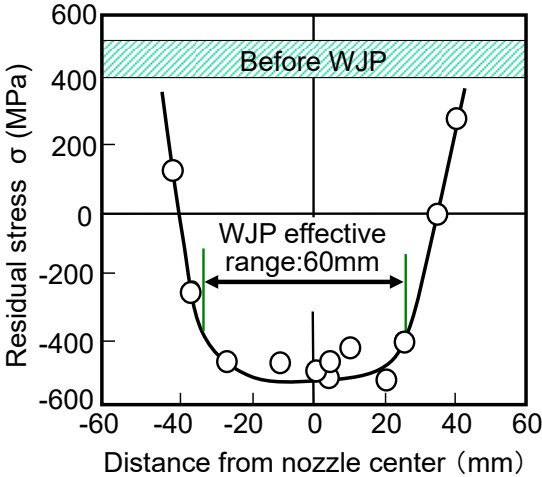

**Figure 4.** Residual stress distribution on a type 304 stainless steel plate surface measured by X-ray diffraction.

The example of the residual stress distribution near surface induced by water jet peening is shown in Figure 5. The residual stresses were measured by sequential polishing X-ray diffraction method. The residual stress on the surface is about −480 MPa. The compressive residual stress gradually decreased with the distance from the surface. The residual stress becomes almost zero at the depth of about 250 μm and turns to tensile. The compressive residual stress depth is shallower than shot peening [13]. However, the depth which the residual stress turns from compressive to tensile can be deeper than 1 mm by controlling the condition of water jet peening.

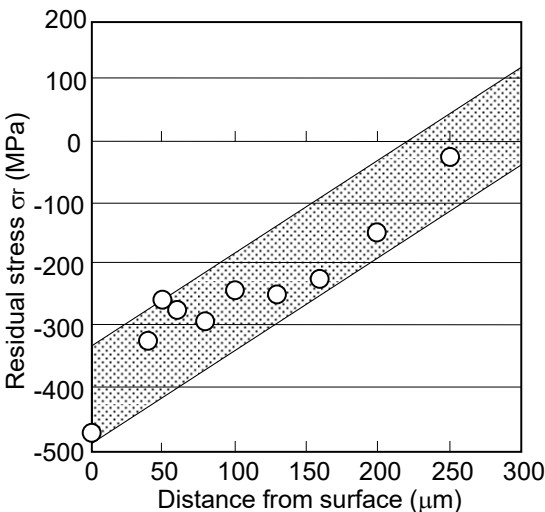

**Figure 5.** Residual stress distribution near surface of type 304 stainless steel plate by water jet peening measured by sequential polishing X-ray diffraction method.

## 3. Pseudo Peak Shift due to Gauge Volume Removal from Sample

Figure 6 shows principle of pseudo peak shift occurrence in neutron diffraction. The neutron beam generated in a nuclear reactor is transferred to a diffractometer through the guide-like so-called supermirror. At the neutron guide side, the neutron is transmitted through double collimator 1 in which the divergence angle is $\alpha1$. It is monochromated by a monochromator. There are many types of monochromator. RESA (Residual Stress Analyzer) in JRR-3 of JAEA employs a bent-type monochromator. It consists of nine Si single-crystal plates. Each single crystal is bent to concentrate

the neutron beam in the horizontal direction and leans lower or higher, and also turns up and down toward the center of the vertical direction to concentrate the neutron beam in the vertical direction. The monochromated neutron beam is transmitted to the measurement sample through collimator 2 in which the divergence angle is $\alpha 2$. The diffracted neutron beam is transmitted to the detector through collimator 3 in which the divergence angle is $\alpha 3$. The gauge volume is determined by the cross section of incident beam and diffraction beam, and its cross-section is not square or circular. Furthermore, wave length depends on the location in the gauge volume. When the gauge volume deviates from the measurement sample for the residual stress measurement of surface layer, the neutron beam is not diffracted from the volume deviating from the sample. In these complicated gauge volume configurations, the wavelength distribution for the gauge volume and the deviation of neutron beam from the sample gives a complicated diffracted beam profile and leads to the generation of pseudo peak shift. It depends on the neutron beam optics and cannot be simply determined theoretically. Thus the pseudo peak shift has to be evaluated with each experiment.

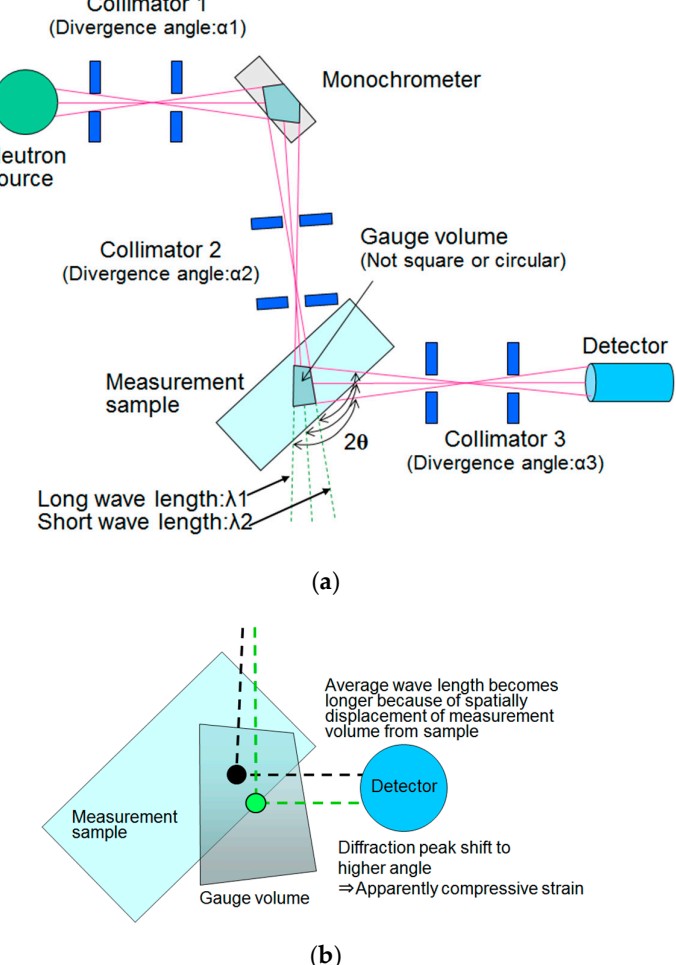

**Figure 6.** Principle of pseudo peak shift occurrence in neutron diffraction. (**a**) Total optics in neutron diffraction; (**b**) optics around gauge volume.

## 4. Experimental Procedure

### 4.1. Specimen

Water jet peened materials are type 304 stainless steel. Chemical compositions and mechanical properties are listed in Tables 1 and 2.

**Table 1.** Chemical compositions (wt%).

| C | Si | Mn | P | S | Cr | Ni | Fe |
|------|------|------|-------|------|-------|------|------|
| 0.07 | 0.44 | 1.18 | 0.032 | 0.01 | 18.26 | 8.32 | Bal. |

**Table 2.** Mechanical properties.

| Yield strength (MPa) | Tensile strength (MPa) | Elongation (%) |
|------|------|------|
| 287 | 640 | 58 |

Shapes and dimensions of measured samples are shown in Figure 7. One is a 300 mm square and 6 mm thick plate specimen. Water jet peening is subjected to a normal plate specimen. The other is bar-like specimen. The cross section is 5 mm wide, 8 mm high and 50 mm long. A notch is introduced on the top surface by the electrical discharge machining, simulating SCC (stress corrosion cracking). Its depth is 3 mm. Water jet peening is used on the top surface.

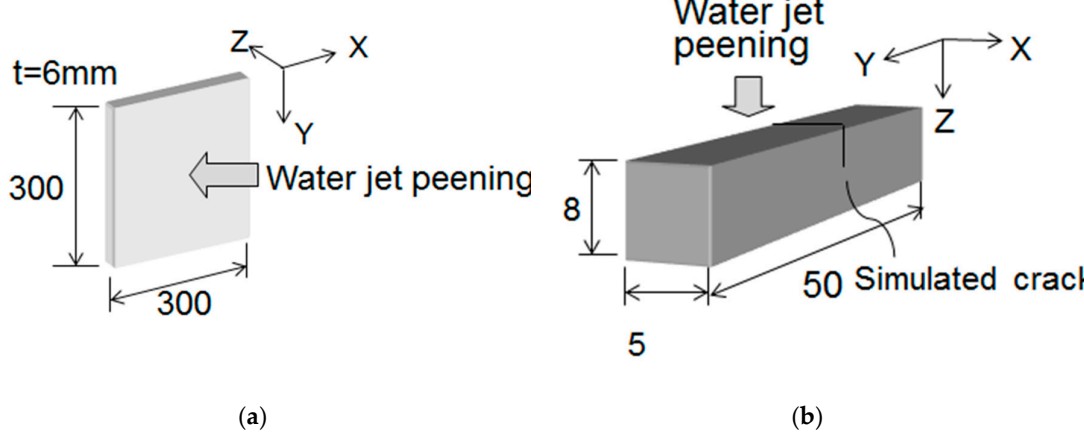

(**a**)                                                                                                     (**b**)

**Figure 7.** Shapes and dimensions of measured samples. (**a**) Plate specimen; (**b**) bar-like specimen with simulated crack.

*4.2. Residual Stress Measurement*

RESA in JRR-3 is shown in Figure 8. The neutron transferred through T2 neutron beam guide from the reactor. It is monochromated by the nine Si single crystal bent monochromator. Wave length is $\lambda = 0.20995$ nm. The monochromated neutron beam is transmitted through incident beam tube and focused by Cd mask. Gauge volumes are $0.5 \times 0.5 \times 13$ mm$^3$ for the plate specimen and $0.3 \times 0.3 \times 8$ mm$^3$ for the bar-like specimen, each defined by the shape of mask. The narrow mask width accommodates a severe residual stress gradient.

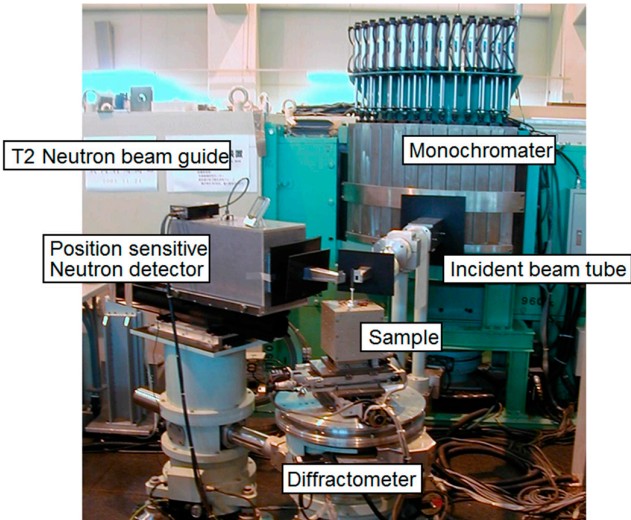

**Figure 8.** Appearance of RESA in JRR-3.

With the three strain measurements at each location, residual stresses can be calculated through a generalized Hook's law,

$$\sigma_\alpha = \frac{E}{(1+v)} [\varepsilon H + \frac{v}{(1-2v)} (\varepsilon H + \varepsilon R + \varepsilon A)] \tag{1}$$

where $E$ is Young's modulus; $v$ is Poisson's ratio; and $\varepsilon_R$, $\varepsilon_A$ and $\varepsilon_H$, are the radial, axial and hoop strain components respectively. The radial, axial and hoop components of stress, $\sigma_R$, $\sigma_A$ and $\sigma_H$ are obtained by cyclic permutation of the indices in Equation (1).

The elastic constants $E$ and $v$ necessary to convert the measured strain to the stress, depend on the hkl diffraction plane. In this experiment the residual strains are measured using 111 diffraction peak. The Young's modulus, $E_{111}$, is 231 GPa and the Poisson's ratio, $v$ is 0.19 for 111 diffraction.

## 5. Results and Discussion

The residual stresses in x, y and z directions in the near-surface of water jet peened type 304 stainless steel plate are shown in Figure 9. In x direction, about 700 MPa tensile residual stress is obtained on the surface; it rapidly decreases with the depth from the surface and turns to compressive at about 1.5 mm. In y direction, the absolute values of the residual stress are very small on the surface; they gradually decease with the depth from the surface and range from 200 to 300 MPa at the depth of about 2 mm. In z direction, the residual stress is about 300 MPa on the surface; it gradually decreases with the depth from the surface, and turns to compressive at about 0.8 mm.

The residual stress distributions in x and y directions were measured by the sequential polishing X-ray diffraction method. The results showed that there was no difference between x and y directions, because the coverage area of the water jet peening as very wide, as shown in Figure 3. The same thing goes for the shot peening. The residual stress distribution in z direction cannot be measured. The comparison of residual stress distributions in water jet peened and shot peened surfaces in x direction are shown in Figure 10. In the figure, open circles reveal the residual stress distribution measured by neutron diffraction in x direction, as shown in Figure 9. Gray circles reveal the residual stress distribution measured by the sequential polishing X-ray diffraction shown in Figure 5. Black circles reveal the residual stress distribution in the shot peened surface layer measured by the sequential polishing X-ray diffraction. It is clear that the residual stress distribution in x direction differs from those measured in shot or water jet peened surface by the sequential polishing X-ray diffraction method.

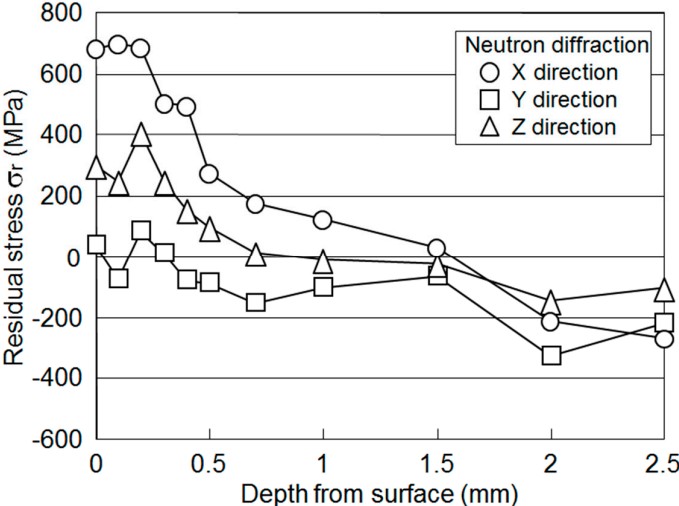

**Figure 9.** Residual stress distribution in near-surface of water jet peened type 304 stainless steel plate.

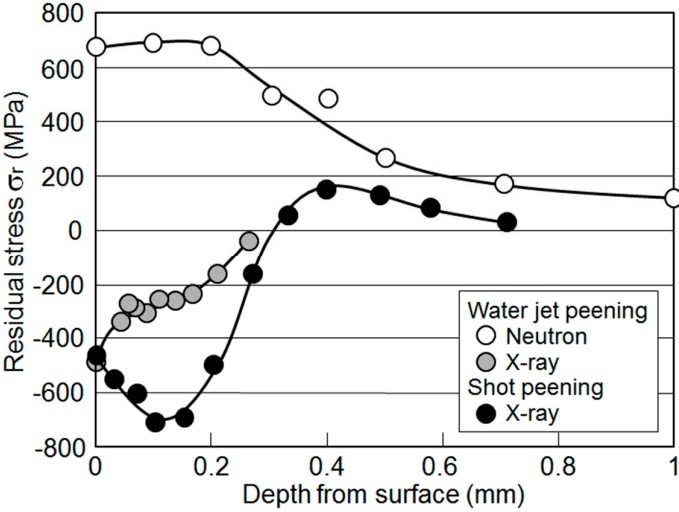

**Figure 10.** Comparison of residual stress distributions.

As mentioned in the introduction, when the gauge volume deviates from the sample, a pseudo peak shift occurs and accurate stress distribution cannot be evaluated. Therefore, the pseudo peak shift has to be measured under the same conditions as in the case of actual residual stress measurement using a sample in an unstressed state. The pseud peak shifts obtained are shown in Figure 11. The pseud peak shifts depend on the diffraction conditions. In the figure the peak shifts for the reflection in z direction and the transmission in x and y directions are shown. The pseud peak shifts on the surface is about 61.99 deg. Since the pseud peak shift disappears when the gauge volume is fully involved in the sample, the diffraction peak angle becomes constant. However it depends on the diffraction condition, so that the pseud peak shift for the transmission disappears at shallower depth and the pseud peak shift for the reflection disappears at deeper depth.

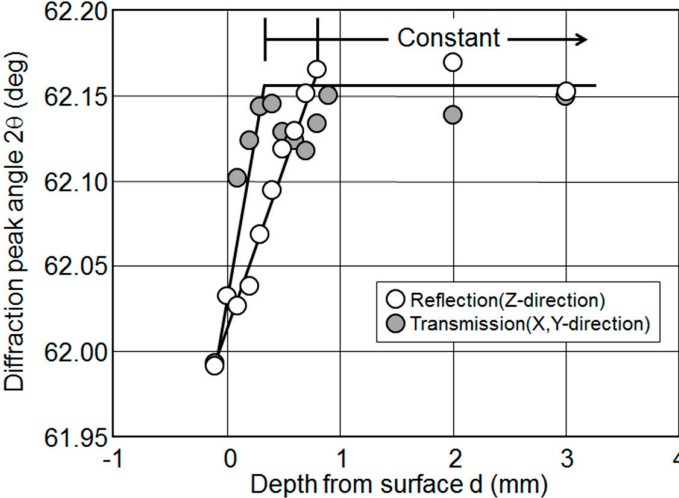

**Figure 11.** Pseudo peak shifts due to gauge volume deviating from measurement sample.

The residual stress distributions in near-surface of water jet peened type 304 stainless steel plate considering the pseud peak shift are shown in Figure 12. In x direction, the residual stress on the surface is about 600 MPa in compression; it rapidly increases with the depth from the surface and turns to tensile at about 0.4 mm. It takes a maximum tensile stress of about 300 MPa at about a depth of 0.9 mm and decreases to zero at the depth of 1.3 mm. In y direction, the residual stress on the surface is about 400 MPa in compression; it rapidly increases with the depth from the surface and turns to tensile at about 0.3 mm. It takes maximum tensile stress of about 200 MPa at about a depth of 0.6 mm and decreases to zero at the depth of 1.3 mm. In order to balance the compressive residual stress near the surface, the residual stress becomes tensile stress from a certain depth. In deeper positions, the residual stress approaches zero because the material is solution heat treated at 1150 °C for 1 h. In z direction the residual stresses range from −150 to 150 MPa from the surface to the depth of 2.2 mm.

The water jet peening gives almost homogeneous residual stress distribution, as shown in Figure 5, so that the residual stress distribution in x and y directions are almost the same. However, there exists about a 200 MPa difference. At present, since the directionality of the water jet peening process does not exist, the difference of residual stress seems to be involved in the measurement accuracy.

The residual stress distribution in x direction measured by the sequential polishing X-ray diffraction method is shown in Figure 12. The measured maximum depth is 0.4 mm. The residual stress distribution in x direction agrees well with that measured by the X-ray.

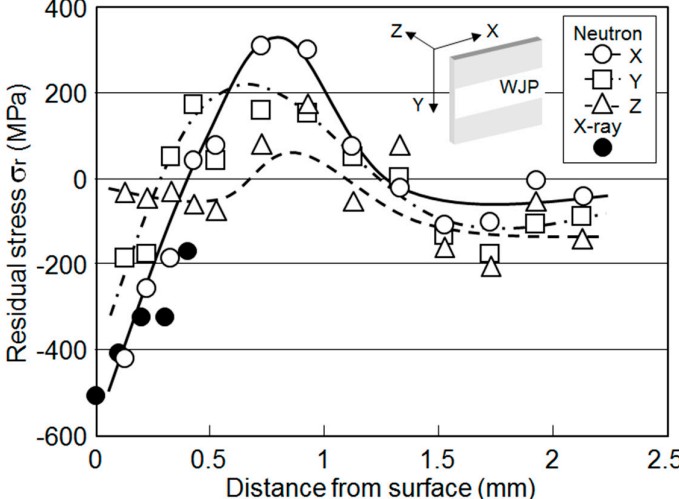

**Figure 12.** Residual stress distributions in near-surface of water jet peened type 304 stainless steel plate considering the pseud peak shift.

The residual stress distributions in near-surface of water jet peened type 304 stainless steel bar with simulated SCC crack considering the pseud peak shift are shown in Figure 13. In x direction, the residual stress on the surface is about 550 MPa in compression; it rapidly increases with the depth from the surface, and close to about zero at the depth of about 0.6 mm, but does not turn to tensile. In y direction, the residual stress on the surface is about 300 MPa in compression; it rapidly increases with the depth from the surface and turns to tensile at about 0.2 mm. It keeps almost zero from the depth of 0.3 to 2.5 mm. In z direction the residual stress on the surface is about 50 MPa in tension, and increases with the depth from the surface. It takes the tensile maximum value at about 0.4 mm and gradually decreases with depth.

The residual stress distribution in x direction measured by the sequential polishing X-ray diffraction method is shown in Figure 13. The residual stress distribution in x direction almost agrees with that measured by the X-ray.

Gray symbols show the residual stress distributions before the water jet peening. In spite of solid solution heat treatment before the peening, the residual stress distributions are little bit scattered in near-surface region. The scattered range is about ±200 MPa. This seems to be caused by the small gauge volume of 0.3 × 0.3 × 8 mm.

In order to improve the measurement accuracy it is necessary to increase the gauge volume or the incident neutron beam intensity. However, the increase of the gauge volume is in contradiction to increase of special resolution for the residual stress distribution at the near-surface with a steep stress gradient. Neutron flux at T-2 thermal neutron guide in JRR-3 is about $1 \times 10^8$ n/cm² s. After monochromation, the flux is reduced to about $1 \times 10^6$ n/cm² s, and the flux at the center of diffractometer is reduced to about $1 \times 10^4$ n/cm² s by passing through incident beam tube. By the introduction of nine Si single anti-symmetry bent crystal monochromators, the flux at the center of diffractmeter is improved to about $5 \times 10^8$ n/cm² s, even though the flux is not sufficient for the gauge volume of 0.3 × 0.3 × 8 mm. The authors think one of the improvements for much higher flux is the stacking of thin Si single crystals. Now the authors are trying to develop such a stacking crystal monochromator structure.

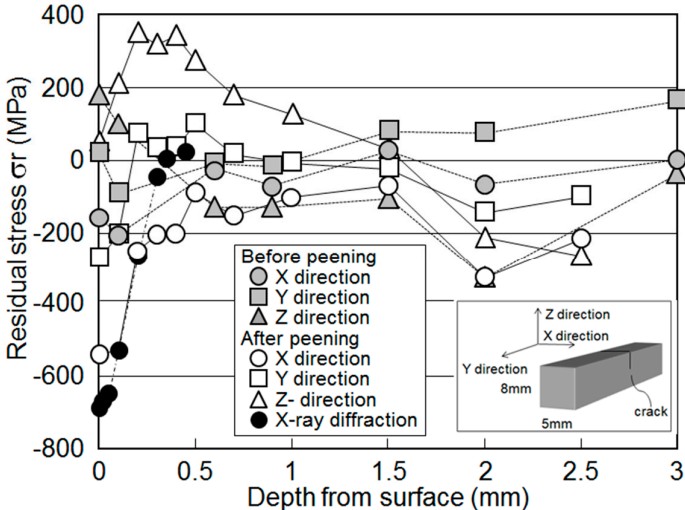

**Figure 13.** Residual stress distributions in near-surface of water jet peened type 304 stainless steel bar with simulated stress corrosion cracking considering the pseud peak shift.

There are two ways to evaluate the residual stress distribution at the near-surface with a steep stress gradient. One is the flipping method. However, the flipping method is hardly applicable to the very thick materials because of limited penetration depth of thermal neutron. The correction method for the pseud peak shift would be much more applicable for thicker materials.

## 6. Conclusions

The residual stress distributions at the near-surface of the water jet peened type 304 stainless steel plate and bar are measured by the neutron diffraction. The conclusions obtained in this study would be summarized as follows:

(1) When the gauge volume deviates from the sample, a pseudo peak shift occurs and accurate stress distributions cannot be evaluated.

(2) If the peak diffraction angles measured for the water jet peened plate and bar samples are corrected using the pseudo peak shift measured in advance under the same conditions as in the case of actual residual stress measurement using a sample in an unstressed state; the residual stress distribution in the surface layer can be evaluated as being in good agreement with the measurement result obtained by the sequential polishing X-ray diffraction method.

(3) The surface layer stress distribution with steep stress gradient can be evaluated even by the neutron diffraction method.

**Author Contributions:** Conceptualization, M.H. and S.O.; methodology, M.H. and H.S.; software, S.O.; validation, M.H., S.O. and H.S.; formal analysis, M.H.; investigation, S.O.; resources, M.H.; data curation, S.O.; writing—original draft preparation, M.H.; writing—review and editing, M.H.; visualization, M.H. and S.O.; supervision, M.H..; project administration, M.H..; funding acquisition, M.H. All authors have read and agreed to the published version of the manuscript.

**Funding:** The authors declare no funding

**Acknowledgments:** The authors express the grateful acknowledgement to Morii, Minakawa and Moriai, Japan Atomic Energy Research Institute (present, Japan Atomic Energy Agency) for supporting the experiment and analysis.

**Conflicts of Interest**: The authors declare no conflict of interest

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
