# Peer review of "Residual Stress Distribution in Water Jet Peened Type 304 Stainless Steel"

_qubs, doi:10.3390/qubs4020018_

Round 1

Reviewer 1 Report

A very interesting article. I miss explaining where the difference in stresses between directions X and Y fig. 9 comes from and description of the method of sequential polishing

Author Response

We observed no difference in the residual stress distributions of shot peened surface by the sequential X-ray diffraction method. And at the first step of the development of water jet peening, the residual stress distribusions in X and Y directions were measured by the sequential polishing X-ray diffraction method and we found no difference in X and Y directions.  Thus we show the residual stress distirbution in X direction in Fig.5 as an example.  

Accordingly we revise the explanation of Fig.9 as "Anyway the residual stress distributions in X and Y directions differ from those measured in shot or water jet peened surface by the sequential polishing X-ray diffraction method."

Reviewer 2 Report

The paper “Residual Stress Distribution in Water Jet Peened Type 304 Stainless Steel” presents an investigation about the stress distributions in the surface layer of type 304 stainless steel subjected to water jet peening. For this purpose, the authors evaluated the residual stress distributions in near-surface of the water jet peened type 304 stainless steel by means of the neutron diffraction.

It presents an adequate introduction, establishing the approach of the problem and the current situation in this field of research.

The equipment and methodology are explained correctly. The results obtained are clear, and maybe the authors should support them with the previous research of other authors. The conclusions adequately summarize the work presented. The presentation and structure of the paper is very correct, so, in my opinion, it can be published with the following minor revision:

  • The reference to the figure in the text and in the captions are incorrect, change Fig. by Figure
  • The sentence in lines 182-183 should be more explained. Maybe it could be appropriate include a new figure comparing the two measurements.
  • Line 206. Why residual stress decreases after the maximum peak? Include reference that support these results.
  • Lines 213-214. “…exist about 200 MPa difference. At the present, since the directionality of water jet peening process does not exist, the difference of residual stress seems to be involved in the measurement accuracy”. Isn't the difference too big to attribute to measurement inaccuracies? Can it be supported with published works?
  • Lines 233-234. “The residual stress distribution in X direction well agrees with that measured by the X-ray”. This statement is not clear in the figure.

Author Response

Comment 1:

 I cannot understand what Reviewer 2 means. I am going to fix what I need to fix.

Comment 2:

According to the comment, I added new Fig.10 and some sentences. Please check the revised file. Line 182-189

Comment 3:

According to the comment, I added some sentences in Line 217-219.

Comment 4:

I think the scattering of residual stress value of about 200 MPa is caused by two reasons. One is the small cross section of gauge volume of 0.5mm x 0.5mm. The other one is the low flux of incident neutron beam. I explain this low flux in Line 250-260.

Comment 5:

According to the comment, I modified the expression. Surely the residual stress distribution does not well agree with that measured by X-ray, it almost agree. Please check Line 244.
